# Formation Mechanism of Carbon-Supported Hollow PtNi Nanoparticles via One-Step Preparations for Use in the Oxygen Reduction Reaction



Dong-gun Kim [1,†], Yeonsun Sohn [1,†], Injoon Jang [2], Sung Jong Yoo [2,3,4,*] and Pil Kim [1,*]

[1] School of Semiconductor and Chemical Engineering, Green Energy Research Center, Jeonbuk National University, Jeonju 54896, Korea; wowkdg@hanmail.net (D.-g.K.); sonys0421@jbnu.ac.kr (Y.S.)
[2] Hydrogen Fuel Cell Research Center, Korea Institute of Science and Technology (KIST), Seoul 02792, Korea; t15759@kist.re.kr
[3] Division of Energy and Environment Technology, KIST School, University of Science and Technology (UST), Daejeon 34113, Korea
[4] KHU-KIST Department of Converging Science and Technology, Kyung Hee University, Seoul 02447, Korea
* Correspondence: ysj@kist.re.kr (S.J.Y.); kimpil1@jbnu.ac.kr (P.K.)
† These authors contributed equally to this work.

**Abstract:** Hollow Pt-based nanoparticles are known to possess the properties of high electrocatalytic activity and durability. Nonetheless, their practical applications as catalytic materials are limited because of the requirement for exhaustive preparation. In this study, we prepared carbon-supported hollow $PtNi_x$ (x = the moles of the Ni precursor to the Pt precursor in the catalyst preparation step) catalysts using a one-step preparation method, which substantially reduced the complexity of the conventional method for preparing hollow Pt-based catalysts. In particular, this hollow structure formation mechanism was proposed based on extensive characterizations. The prepared catalysts were examined to determine if they could be used as electrocatalysts for the oxygen reduction reaction (ORR). Among the investigated catalysts, the acid-treated hollow $PtNi_3$/C catalyst demonstrated the best ORR activity, which was 3 times higher and 2.3 times higher than those of the commercial Pt/C and acid-treated particulate $PtNi_3$/C catalysts, respectively.

**Keywords:** Pt-based catalysts; hollow PtNi alloy nanoparticles; galvanic displacement reaction; oxygen reduction reaction



## 1. Introduction

Hollow materials have been identified as promising candidates for adsorption and catalysis because of their unique structural characteristics, such as high stability and high surface area, as well as their physical and chemical properties [1–3]. When they are composed of metals, the arrangement of the surface atoms differs from that of the dense structure, resulting in changes in the electron structure and adsorption behavior of reactant molecules [4,5]. These interesting surface properties, in combination with a high mass-based surface area, make hollow metal particles suitable as high-performance catalysts [6–8].

Hollow particles have been prepared using a template made of spherical silica or polymer; this method involves the formation of a target material on the template surface, followed by the selective removal of the template [9–11]. Some critical disadvantages of this method include the difficulty of reducing the size of the hollow particles because of the large template, resulting in a limited enhancement in catalytic performance. The large size of the template tends to form thick hollow shells that are usually composed of small particle aggregates, causing the effect of the hollow cage on the strain of the surface atom to be limited [12]. This demonstrates the importance of shell surface integrity in terms of the adsorption energy for catalytic processes. Therefore, hollow metal particles

should have properties such as small sizes and smooth surfaces in order to be suitable for catalytic applications [13,14].

Hollow precious metal particles have also been prepared using galvanic displacement, which uses the large difference in the standard potential between two metals [15–17]. A sacrificial metal template was made before the reaction with the target metal precursor, and eventually, hollow metal particles were formed spontaneously as electrons directly transferred from the metal template to the target metal ions. Small-sized hollow metal particles with smooth surfaces were able to be produced because the size of the metal template could be controlled to several nanometers, and the resulting hollow metal nanoparticles have been reported to have high catalytic performance [18–20]. Nonetheless, galvanic displacement should not be used for the mass production of hollow metal nanoparticles because of its highly constrained synthesis conditions. The sacrificial metal templates, which have relatively low standard potentials, are prepared in an air-exclusive environment prior to the displacement reaction with the target metal ions unless no hollow structure is formed [21–23]. In addition, the surfactants used for stabilizing the metal template must be removed before the prepared hollow metal particles for catalytic applications [24]. Our group developed a hollow metal nanoparticle synthesis method that avoids these difficult preparation conditions. Using this method, carbon-supported hollow PtNi nanoparticles were formed in a single step under conventional conditions for the co-impregnation of binary metals. The prepared hollow PtNi nanoparticles exhibited high catalytic performance in the oxygen reduction reaction (ORR). However, the one-step formation mechanism of PtNi hollow nanoparticles has not yet been thoroughly understood, even though understanding it is essential for the scaling up of hollow nanoparticle manufacturing.

In this study, carbon-supported hollow PtNi nanoparticles with various compositions were prepared using a one-step method under various synthesis conditions in order to investigate the hollow structure formation mechanism. The prepared hollow nanoparticles were analyzed using various characterization techniques, such as X-ray diffraction (XRD), transmission electron microscopy (TEM), inductively coupled plasma optical emission spectroscopy (ICP-OES), and X-ray photoelectron spectroscopy (XPS). Then they were used as electrocatalysts for the ORR. In particular, the reaction-time dependent morphology of hollow PtNi nanoparticles was intensively characterized in order to propose the formation mechanism of PtNi hollow nanoparticles. It was found that the Ni–Pt core-shell nanoparticles were first generated by the reduction of metal precursors; PtNi hollow structures were then developed by the Kirkendall effect [25–29].

## 2. Results and Discussion

Figure 1 shows the HR-TEM images and line-scan profiles of $PtNi_3/C$–H, $PtNi_3/C$–S, $PtNi_3/C$–H AT, and $PtNi_3/C$–S AT. Both $PtNi_3/C$–S and $PtNi_3/C$–S AT exhibit the typical morphology of small particle aggregates, which is common for bimetallic particles that have not been treated at high temperatures. A slight change in the line-scan profile of Ni can be observed because of the dissolution of Ni during acid treatment. In contrast to these samples, both $PtNi_3/C$–H and $PtNi_3/C$–H AT have hollow structures. This is obvious in their elemental line-scan profiles, which show double-humped intensities for the Pt element on the particle edges, indicating that a Pt shell structure was formed. The line-scan profiles of Ni are similar to those of Pt, but they are less visible in $PtNi_3/C$–H AT than in $PtNi_3/C$–H, implying that some Ni was dissolved out during the acid washing of $PtNi_3/C$–H. It is worth noting that the hollow structure remains unchanged before and after acid washing, indicating that, in addition to its high stability, its formation was due to a true one-step reaction rather than the acid leaching of Ni.

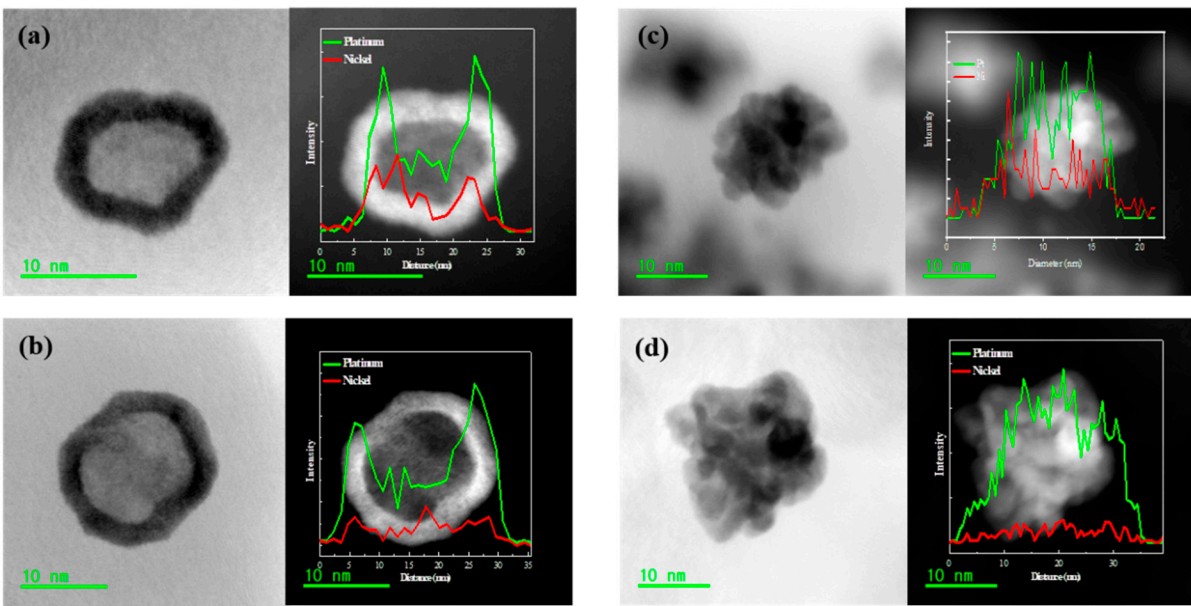

**Figure 1.** High-resolution (HR)-TEM images and line-scan profiles of (**a**) PtNi$_3$/C–H, (**b**) PtNi$_3$/C–H AT, (**c**) PtNi$_3$/C–S, and (**d**) PtNi$_3$/C–S AT.

The surface states of Pt and Ni were characterized by XPS. Figure 2 shows the XPS spectra of PtNi$_3$/C–H, PtNi$_3$/C–S, PtNi$_3$/C–H AT, and PtNi$_3$/C–S AT. All the samples reveal two peaks at 70.8 eV and 74.2 eV, which correspond to the photoelectron peaks of Pt 4f$_{5/2}$ and Pt 4f$_{7/2}$, respectively (Figure 2a). The intensities of the Pt photoelectron peaks of the acid-treated samples are stronger than those of the untreated samples, indicating that Ni on the catalyst surfaces was removed by acid washing. This is confirmed by the Ni photoelectron spectrum (Figure 2b), which shows that the characteristic Ni peaks disappeared after acid treatment. The intensities of the Pt photoelectron peaks for PtNi$_3$/C–H and PtNi$_3$/C–S are noteworthy. PtNi$_3$/C–S displays weaker intensities of Pt 4f$_{5/2}$ and Pt 4f$_{7/2}$ than PtNi$_3$/C–H, indicating that its Pt surface concentration is significantly lower. In PtNi$_3$/C–S, the reduction rate of the Pt ions is higher than that of the Ni ions. When these two metal precursors are reduced simultaneously, a large portion of Pt is deposited on the core of the particles, while Ni tends to be on the surface, resulting in a weak intensity of Pt in the XPS spectrum. Consequently, the Pt precursor for PtNi$_3$/C–H in our study had a lower reduction rate than the Ni precursor, resulting in a higher Pt surface concentration.

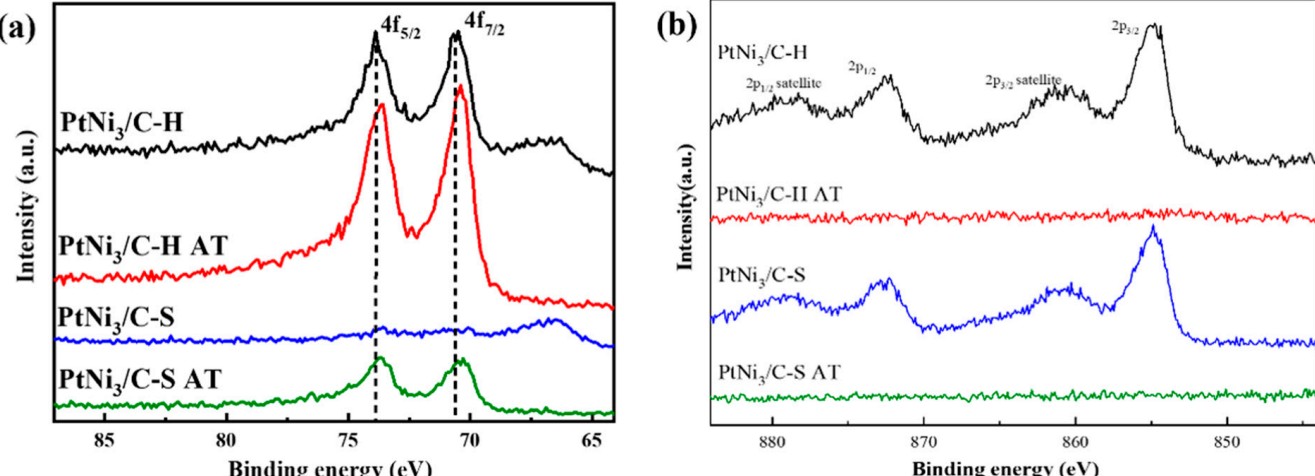

**Figure 2.** (**a**) Pt 4f and (**b**) Ni 2p XPS spectra of catalysts.

Figure 3 shows the TEM images of the acid-treated samples. PtNi$_3$/C–S (Figure S1) has irregular particle aggregates with film-like structures around them, which were then removed by acid treatment, as evidenced by Figure 3d. However, there is a negligible difference in the morphology between PtNi$_x$/C–H and PtNi$_x$/C–H AT, indicating that the hollow structures are stable regardless of the catalyst composition. As the ratio of Ni to Pt increased, the cavity size became large (Figure S1), demonstrating that the Ni content in the preparation medium is an important factor in determining the cavity size.

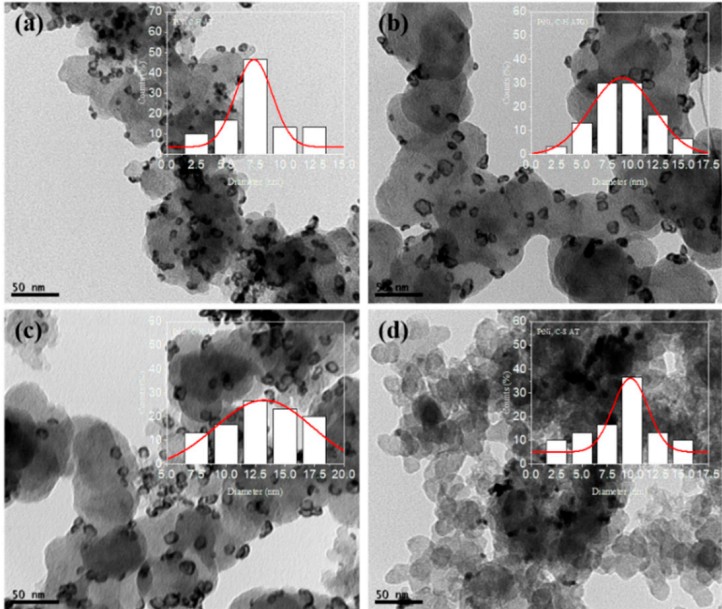

**Figure 3.** TEM images of (**a**) PtNi/C–H AT, (**b**) PtNi$_2$/C–H AT, (**c**) PtNi$_3$/C–H AT, and (**d**) PtNi$_3$/C–S AT.

Table 1 summarizes the metal contents of the prepared catalysts. PtNi$_3$/C–H and PtNi$_3$/C–S have similar compositions, implying that most of the metal components added to the preparation solution were supported on carbon regardless of the type of Pt precursor. Large amounts of Ni were removed from the catalyst after acid treatment, as confirmed by the compositions of PtNi$_3$/C–H AT and PtNi$_3$/C–S AT. The Ni contents in the PtNi$_x$/C–H AT catalysts slightly increased as the Ni to Pt precursor ratio increased during the preparation step. This is obvious considering the Pt/Ni values for PtNi$_3$/C–H and PtNi$_3$/C–S (Table 1). Although the Pt/Ni ratio measured by ICP-OES was identical in these two catalysts, they revealed large differences in the XPS-derived Pt/Ni ratio. PtNi$_3$/C–H had a 6.3 times higher Pt/Ni ratio than PtNi$_3$/C–S, indicating a higher surface concentration of Pt in PtNi$_3$/C–H.

**Table 1.** Metal content of the catalyst measured by ICP-OES.

|  | Pt Contents (wt%) | Ni Contents (wt%) | Atomic Ratio (Pt/Ni) [1] | Atomic Ratio (Pt/Ni) [2] |
| --- | --- | --- | --- | --- |
| Pt/C commercial | 19.6 | - | - | - |
| PtNi/C–H AT | 22.6 | 0.70 | 9.7 | - |
| PtNi$_2$/C–H AT | 19.1 | 0.73 | 7.9 | - |
| PtNi$_3$/C–H AT | 15.8 | 0.75 | 6.3 | 5.6 |
| PtNi$_3$/C–S AT | 15.7 | 1.52 | 3.1 | 2.1 |
| PtNi$_3$/C–H | 15.9 | 13.9 | 0.3 | 0.82 |
| PtNi$_3$/C–S | 15.8 | 14.0 | 0.3 | 0.13 |

[1] Measured by ICP-OES; [2] Measured by XPS.

Figure 4 presents the XRD patterns of the acid-treated samples and commercial Pt/C catalyst. Although the peak positions vary with the catalyst composition, all samples exhibit a similar diffraction pattern. For example, Pt/C displays the characteristic Pt (111), (200), and (220) peaks at 39.6°, 47.4°, and 67.1°, respectively [30,31]. Compared to Pt/C, $PtNi_x$/C–H AT and $PtNi_3$/C–S AT have diffraction peaks that are shifted to higher angles; the degree of angle shift is dependent on the catalyst composition. As the Ni to Pt ratio increases, the diffraction peaks shift to higher angles, which is due to the lattice contraction of Pt, resulting from the small size of Ni compared to that of Pt. This is evident from the lattice constant presented in Table 2, which shows that the lattice constants decreased as the relative contents of Ni versus Pt increased. It is worth noting that $PtNi_3$/C–H AT has a smaller lattice constant than $PtNi_3$/C–S AT even though the former has a lower Ni concentration. This increased lattice contraction of $PtNi_3$/C–H AT is likely due to the hollow structure-induced effect on the lattice strain. A similar lattice contraction trend was also observed in the samples before acid treatment (Figure S2). $PtNi_x$/C–H and $PtNi_3$/C–S have additional peaks at 33.5° and 59.7°, which can be assigned to Ni hydroxide that was dissolved away by acid treatment [32,33]. The crystal sizes of $PtNi_x$/C–H AT and $PtNi_3$/C-S AT were calculated using (220) diffractions. The calculated crystal size of $PtNi_x$/C–H AT corresponds to the shell thickness, while that of $PtNi_3$/C–S AT corresponds to the average particle size. As shown in Table 2, the particle size of $PtNi_3$/C–S AT was calculated to be 5.3 nm. The degree of variation in the shell thickness of $PtNi_x$/C–H AT is negligible, but it did slightly increase as the relative contents of Ni to Pt increased. This is because the size of the hollow cavity increased as the relative concentration of the Ni precursor to the Pt precursor increased in the synthesis medium, as confirmed by TEM.

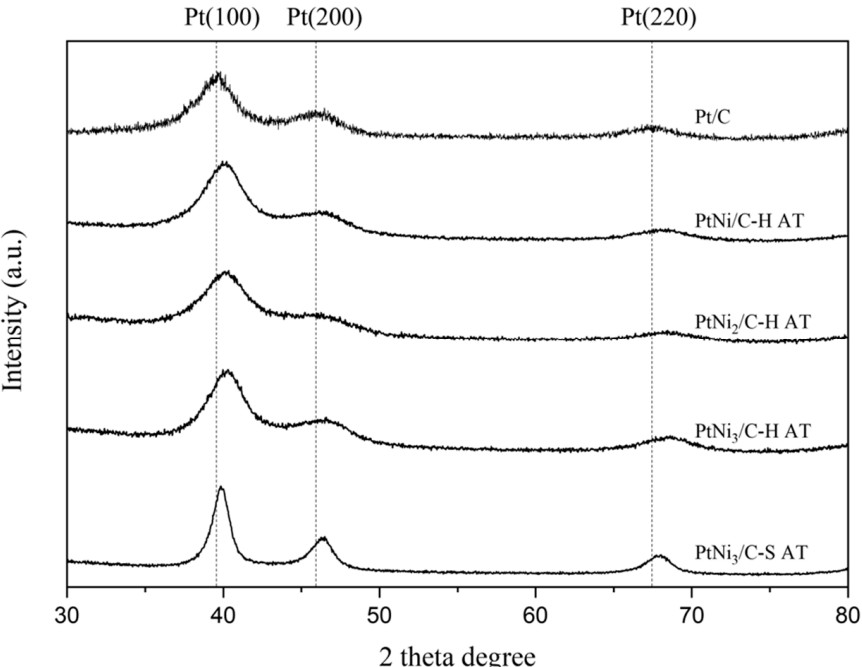

**Figure 4.** XRD pattern of $PtNi_x$/C AT.

The reaction time-dependent morphology was analyzed to investigate the hollow structure formation mechanism. Figure 5 shows the HR-TEM images and element line-scan profiles of $PtNi_3$/C–H taken at different reaction times. Regardless of the reaction time, all the samples shown have a similar morphology, but their element line-scan profiles differ slightly. $PtNi_3$/C–H at 0 min, which was the sample obtained immediately after the introduction of reducing agent had been completed, has a core-shell structure, as confirmed by the EDS line-scan profile. The intensity for Pt shows two peaks centered on the particle edges, while that for Ni reveals a single broad profile heightened on the core. As the

reaction time progressed, there was a negligible change in the shape of the Pt profile, but the element line-scan profile of Ni began to resemble that of Pt. This implies that Ni in the center was diffused out, resulting in a hollow structure.

**Table 2.** Particle average size, shell thickness, and lattice constant of PtNi$_x$/C AT.

|  | PtNi/C–H AT | PtNi$_2$/C–H AT | PtNi$_3$/C–H AT | PtNi$_3$/C–S AT | Pt/C |
|---|---|---|---|---|---|
| Average size (nm) [1] | 7.6 | 9.3 | 13.0 | 6.1 | - |
| Shell thickness or crystallite size (nm) [2] | 2.6 | 2.7 | 2.8 | 5.3 [3] | - |
| Pt(111) Lattice constant (Å) | 3.91 | 3.89 | 3.87 | 3.92 | 3.92 |

[1] Determined by TEM image; [2] Calculated by Scherrer's equation using Pt(220) diffraction; [3] Crystal size.

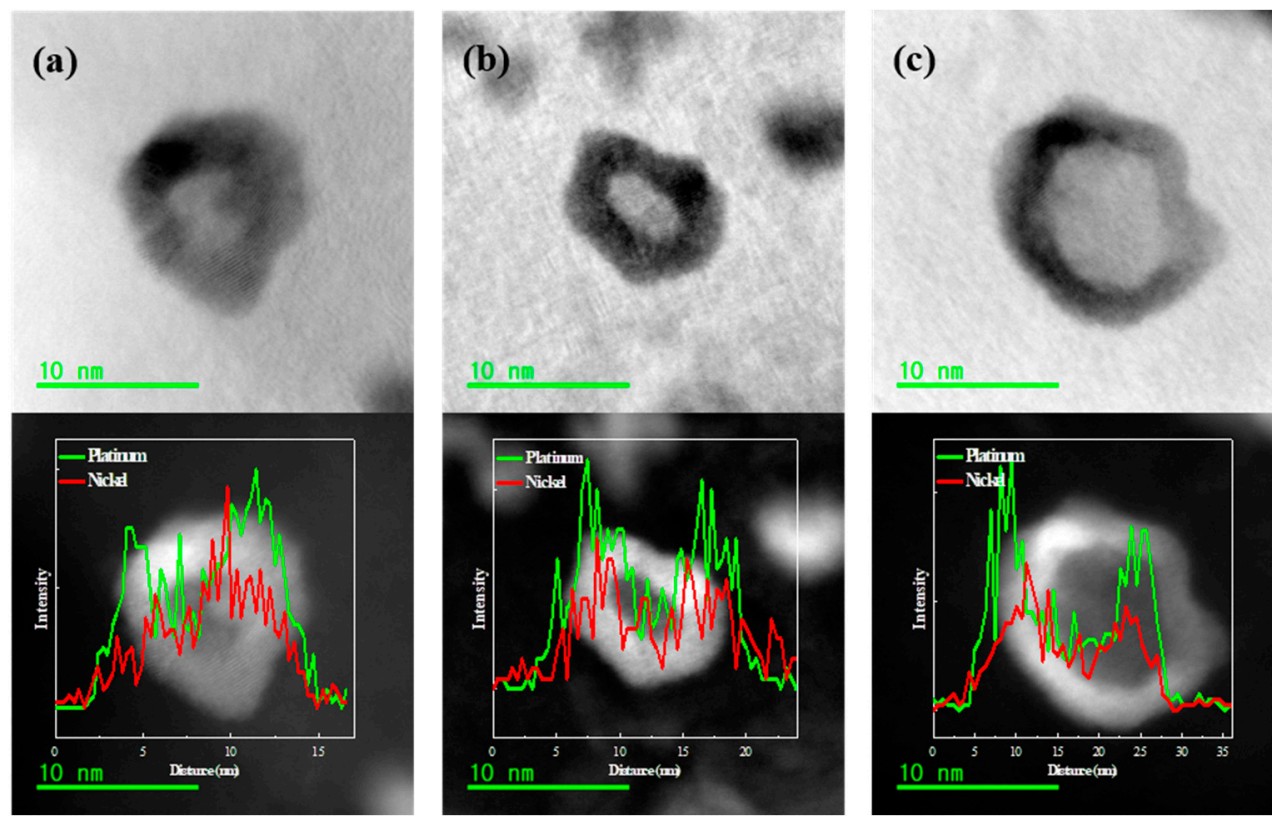

**Figure 5.** HR-TEM images and line-scan profiles of PtNi$_3$/C-H at (**a**) 0 min, (**b**) 15 min, and (**c**) 30 min.

We propose the hollow structure formation mechanism of PtNi$_x$/C–H based on the aforementioned findings (Figure 6). The key factor in the development of hollow structures is the reduction rate of the Pt precursor. In the case of PtNi$_3$/C–S, the Pt ions formed by dissolving the H$_2$PtCl$_6$ precursor had a higher reduction potential than the Ni ions. Therefore, when these two types of ions are simultaneously reduced by a chemical reducing agent such as NaBH$_4$, dense particles consisting of Pt and Ni species are produced. Conversely, in the case of PtNi$_x$/C–H, the Pt ions generated by dissolving the Pt(NH$_3$)$_4$Cl$_2$ precursor had a lower reduction rate than those generated by dissolving H$_2$PtCl$_6$. Thus, the particle formation mechanism used for the PtNi$_3$/C–S catalyst cannot be used for the PtNi$_x$/C–H catalysts. When the reducing agent was added to the solution containing both Pt(NH$_3$)$_4$Cl$_2$ and Ni precursors, a relatively large amount of Ni, as compared to Pt, was produced because of the lower reduction rate of the Pt ions derived from Pt(NH$_3$)$_4$Cl$_2$, resulting in the formation of Ni-rich cores. Subsequently, the Pt ions were reduced on the core surfaces because of the catalytic action of the Ni-rich cores, resulting in the formation of Ni-rich cores and Pt-rich shell structures, as shown in Figure 5a. After the formation

of the core-shell particles, the hollow structure was developed by the Kirkendall effect, in which the difference in diffusion rate between Pt and Ni generates cavities inside the particles, as shown in Figure 5b,c [34,35]. According to the proposed formation mechanism, the cavity size is considered to be highly affected by the Ni precursor concentration, and it determines the size of the Ni-rich cores. For example, Ni-rich cores with large sizes tend to be generated when the Ni precursor concentration is high, resulting in the formation of hollow particles with large cavities. This explains the trend in the PtNi$_x$/C–H cavity size.

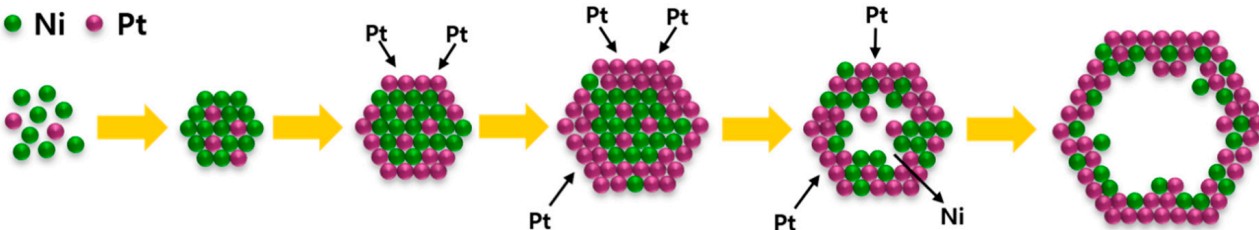

**Figure 6.** Schematics of the hollow PtNi nanoparticles.

The prepared carbon-supported hollow PtNi catalysts were examined for their potential application in the ORR. Figure 7a shows the CVs of the acid-treated samples and commercial Pt/C measured in an N$_2$-saturated acid electrolyte. All the examined catalysts are shown with CV curves that are similar to that of a typical Pt-based catalyst. The electrochemical active surface area (EASA) was calculated using a charge for hydrogen adsorption. As shown in Table S3, the PtNi$_x$/C–H AT catalysts had a higher EASA than the PtNi$_3$/C–S AT catalysts, indicating that the hollow structure had a positive effect on the EASA. The EASA of PtNi$_x$/C–H AT increased as the Ni to Pt ratio increased. The samples before acid-treatment exhibited the same CV curves and EASA trends as the acid-treated samples (Figure S3 and Table S4). The EASA of the catalysts was found to have increased after acid treatment, indicating that the Ni species on the Pt surface was removed. The ORR performance of the acid-treated samples was evaluated based on the polarization curve measured in an O$_2$-saturated electrolyte (Figure 7b). All the prepared samples had higher half-wave potentials than commercial Pt/C. In particular, PtNi$_x$/C–H AT had higher half-wave potentials than PtNi$_3$/C–S AT, indicating that the hollow PtNi catalysts had better ORR performance. This was confirmed by considering the calculated kinetic current. As shown in Figure S4, the PtNi$_x$/C–H AT catalysts had higher mass-specific kinetic currents than did the Pt/C and PtNi$_3$/C–S AT catalysts. In the case of the PtNi$_x$/C–H AT catalysts, the Ni concentration in the structure increased as the kinetic current increased. Both the EASA and the surface strain are known to influence the ORR performance of Pt-based catalysts. PtNi$_3$/C–H AT had the highest EASA and the smallest lattice constant (the largest surface strain) among the examined catalysts, resulting in it having the largest active area that was not poisoned by oxidized species and consequently, the highest ORR kinetic current. PtNi$_3$/C–H AT had a mass activity of 0.684 A/mg$_{Pt}$, which is 3 times higher and 2.3 times higher than those of commercial Pt/C and PtNi$_3$/C–S AT, respectively. Electrocatalyst durability is a critical factor in determining the lifetimes of fuel cells. We conducted an accelerated durability test (ADT) on the best-performing catalyst, PtNi$_3$/C–H AT, and on the Pt/C catalyst. Figure S5 shows CV and LSV before and after the ADT. PtNi$_3$/C–H AT and Pt/C had EASA degradation percentages of 24.7% and 30.1%, respectively. PtNi$_3$/C–H AT and Pt/C also had mass activities after the ADT of 0.482 A/mg$_{Pt}$ and 0.176 A/mg$_{Pt}$, respectively (Table S5). This demonstrates the high stability of PtNi$_3$/C–H AT under ORR conditions. For the purpose of comparison, the commercial Pt/C catalyst (Premetek) was characterized under the same conditions as those used for the prepared catalysts.

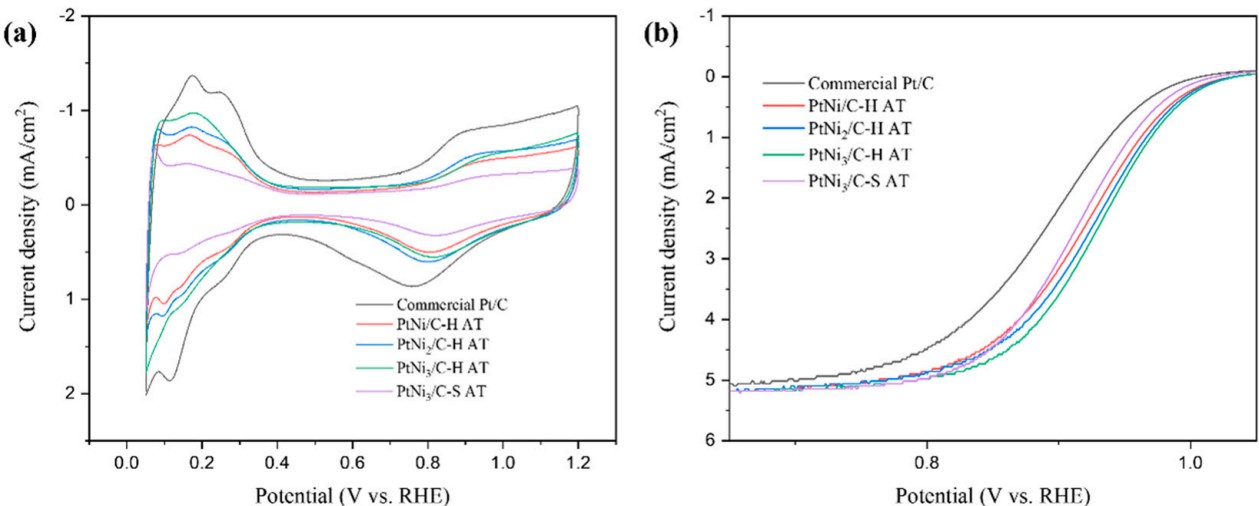

**Figure 7.** (**a**) CV and (**b**) ORR LSV of the catalysts after acid treatment.

## 3. Materials and Methods

### 3.1. Catalyst preparation

Carbon-supported hollow PtNi nanoparticles were prepared using a conventional co-impregnation method. Briefly, $Pt(NH_3)_4Cl_2$ (Alfa Aesar, 98%, US) and $NiCl_2$ (Sigma-Aldrich, 98%, St. Louis, MO, USA) were dissolved in a solution (140 mL of water and 10 mL of ethanol) containing a commercial carbon (XC-72R Vulcan, Cabot, Cabot Corp., Boston, MA, USA). A 0.21 M aqueous $NaBH_4$ solution (Sigma-Aldrich, 98%, USA) was added dropwise to the abovementioned mixture of metal precursors and carbon support under vigorous stirring. After 1 h of reaction, the solid component was separated by filtering and washing with tertiary distilled water (18.2 M$\Omega$). The resultant substance was dried in an oven at 100 °C for 12 h to obtain $PtNi_x/C$–H (C and H stand for the carbon support and the hollow structure, respectively). To study the synthesis time-dependent morphology, samples were obtained from the preparation solution at 0, 15, and 30 min after the $NaBH_4$ solution had been added and subsequently washed with water to prevent the residual metal ions from reacting further. For comparison, particulate $PtNi_3/C$–S (S stands for solid particle) was prepared using the same method as that used for $PtNi_3/C$–H, except that $H_2PtCl_6$ (Kojima, Okayama, Japan) was used as the Pt precursor. The prepared $PtNi_x/C$-H and $PtNi_3/C$–S catalysts were treated with a 3 M HCl solution for 30 min, washed with water, and dried in an oven for 12 h to obtain $PtNi_x/C$–H AT and $PtNi_3/C$–S AT (where AT stands for acid treatment).

### 3.2. Characterizations

An XRD pattern was obtained using Rigaku's MAX-2500 (Cu-K$\alpha$ radiation, 40 kV 30 mA, Rigaku, Akishima, Japan). The sample morphology was studied using a transmission electron microscope (HITACHI H-7650 Hitachi Ltd., Tokyo, Japan and JEM-2010, JEOL Ltd., Tokyo, Japan). The metal contents were determined using ICP-OES (Optima 5300 DV, Perkin Elmer, Waltham, MA, USA). The oxidation state of the surface species was characterized using XPS (Nexsa XPS system, Thermo Fisher, Waltham, MA, USA).

Electrochemical analyses were conducted in a cell that contained three electrodes: working (rotating ring disk electrode, RRDE, Pine Research Instrumentation Inc., Durham, NC, USA), counter (carbon rod), and reference (Ag/AgCl (3 M Cl$^-$ saturated)) electrodes. The working electrode was prepared by being coated with the catalyst ink. The catalyst ink was prepared using a previously reported method [36]. Cyclic voltammetry was carried out in an $N_2$-saturated 0.1 M $HClO_4$ solution under a static condition (scan rate: 100 mV/s and potential range: 0.05–1.2 V (vs. RHE)). Linear sweep voltammetry was conducted in an $O_2$-saturated 0.1 M $HClO_4$ solution (scan rate: 10 mV/s and potential range: 0.05–1.15 V (vs.

RHE)) under the condition of a rotating electrode (rotating speed: 1600 rpm). The catalyst loading on the working electrode was 16 $\mu g_{Pt}/cm^2$. To examine the durability of the catalyst, 10,000 potential cycles in the potential range of 0.6–1.1 V were conducted in an $O_2$-saturated electrolyte before cyclic voltammograms (CVs) and linear sweep voltammograms (LSVs) were measured.

## 4. Conclusions

In this study, carbon-supported hollow PtNi nanoparticles were prepared using a one-step method, where a conventional co-impregnation of Pt and Ni was effective for the formation of hollow structures when $Pt(NH_3)_4Cl_2$ rather than $H_2PtCl_6$ was used as a Pt precursor. This hollow structure formation mechanism was proposed based on the physical characterizations, particularly the preparation time-dependent morphology with element distribution throughout the particle. Since the reduction rate of the Pt ions derived from $Pt(NH_3)_4Cl_2$ was relatively slow, Ni-rich cores and Pt-rich shell structures were generated immediately after the reducing agent was added. These core-shell nanoparticles underwent a diffusion-related reaction known as the Kirkendall process, forming the final hollow structure. The prepared catalysts were used as electrocatalysts for the ORR. The catalyst $PtNi_x/C$–H AT exhibited better ORR performance than did $PtNi_3/C$–S AT. Among the examined catalysts, $PtNi_3/C$–H AT had the highest ORR activity, which was 3 and 2.3 times higher than that of commercial Pt/C and $PtNi_3/C$–S AT, respectively. In addition, $PtNi_3/C$–H AT was found to be more durable than Pt/C in that its mass activity after the AST was still higher than the initial mass activity of Pt/C.

**Supplementary Materials:** The following supporting information can be downloaded at: https://www.mdpi.com/article/10.3390/catal12050513/s1, Table S1: Particle size of $PtNi_x/C$; Table S2: Lattice constant of $PtNi_x/C$; Table S3: Electrochemical surface area, mass activity and specific activity of catalysts; Table S4: EASA of commercial Pt/C and $PtNi_3/C$-H AT at 0.9 V (vs. RHE) after 10,000 cycle ADT; Table S5: Mass activity of commercial Pt/C and $PtNi_3/C$-H AT at 0.9V(vs. RHE) after 10,000 cycle ADT; Figure S1: TEM images of (a) PtNi/C-H, (b) $PtNi_2/C$-H, (c) $PtNi_3/C$-H, (d) $PtNi_3/C$-S; Figure S2: XRD pattern of $PtNi_x/C$; Figure S3: (a) CV and (b) ORR LSV of catalysts before acid treatment; Figure S4: (a) EASA and (b) ORR Mass activity of catalysts; Figure S5: (a,c) Cyclic voltammogram and (b,d) ORR LSV of commercial Pt/C, $PtNi_3/C$-H AT before and after 10,000 cycles ADT.

**Author Contributions:** Conceptualization, Y.S. and P.K.; formal analysis, I.J.; data curation, Y.S. and D.-g.K.; writing—original draft preparation, D.-g.K. and Y.S.; writing—review and editing, D.-g.K. and P.K.; supervision, P.K. and S.J.Y. All authors have read and agreed to the published version of the manuscript.

**Funding:** This research was funded by [NRF of Korea grant] grant number [2018M1A2A2061975, 2020M3H4A3106313, 2021M3H4A1A02042948].

**Data Availability Statement:** Data sharing is not applicable to this article.

**Acknowledgments:** We acknowledge the support of the high-quality TEM analysis from the center for University-Wide Research Facilities (CURF) at Chonbuk National University.

**Conflicts of Interest:** The authors declare no conflict of interest.

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
