# Peer review of "Formation Mechanism of Carbon-Supported Hollow PtNi Nanoparticles via One-Step Preparations for Use in the Oxygen Reduction Reaction"

_catalysts, doi:10.3390/catal12050513_

Round 1

Reviewer 1 Report

The article "Formation Mechanism of Carbon-supported Hollow PtNi Nanoparticles via One-step Preparations for Use in the Oxygen Reduction Reaction" is devoted to the synthesis of hollow nanoparticles with high catalytic activity and stability. Despite their relatively large size and low Ni content, after treatment in acid, these materials show very high catalytic activity and stability compared to Pt/C. The composition and structure of the obtained materials were detail studied by modern methods: XRD, XPS, HRTEM, ICP-OES. However, there are a few things to note about this article:

The literature review presents similar works on the synthesis of hollow nanoparticles. It is necessary to clearly indicate what is the novelty of this particular work. In addition, recent publications on this topic should be added.

section 2.1 does not specify the catalyst drying temperature.

section 2.2 does not provide a reference to the previously described method for preparing catalytic ink

It is not specified what brand of Pt/C catalyst was used for comparison.

According to TEM data, it is necessary to determine the average size of nanoparticles and imagine a histogram of the particles size distribution. How different are the sizes of hollow particles, are there ordinary small nanoparticles?

The article presents XPS data, but these results do not calculate the Pt-Ni atomic ratio for the obtained samples. This information needs to be added. Are there any differences in the material compositions under the XPS and ICP-OES data?

Reviewer 2 Report

Title: Formation Mechanism of Carbon-supported Hollow PtNi Nanoparticles via One-step Preparations for Use in the Oxygen Reduction Reaction

Authors: Dong-gun Kim, Yeonsun Sohn, Injoon Jang, Sung Jong Yoo, Pil Kim

General Comments:

  • In this paper, carbon-supported hollow PtNix (x = the moles of the Ni precursor to the Pt precursor in the catalyst preparation step) catalysts were prepared.
  • The structure of the article fulfills partially the structure of a research article. The section 2. Experimental, should be rename Materials and Methods, and placed after the section Results and Discussion, according to the journal requirements.
  • Four keywords are included by the authors.
  • The Introduction section provide sufficient background information for readers in the immediate field to understand the problem that this study addresses.
  • The authors present in the Experimental section shortly the reagents, the experimental method and the equipment used. The methods and protocols are described in sufficient detail in order to allow another researcher to reproduce the results.
  • In the Results and Discussion section, the authors present and interpret the results of the performed experiments.
  • The paper ends with the Conclusions part. In this section the authors mention the conclusions of their research study.

I suggest to Reconsider after Major Revisions for the following reasons:

  1. pg.1, Abstract, PtNi3/C, “3”should be subscript;
  2. pg. 2, the sentence: “In addition, surfactants have been used for stabilizing the metal template, which must be removed before the prepared hollow metal particles can be used for catalytic applications.” Should be reformulated. One possible reformulation is: “In addition, surfactants have been used for stabilizing the metal template, which must be removed before using the prepared hollow metal particles can be used for catalytic applications”;
  3. The section 2. Experimental, should be rename Materials and Methods, and placed after the section Results and Discussion, according to the journal requirements.
  4. pg. 2: the authors should explain what “S” from the PtNi3/C-S means;
  5. pg. 3: Results and discussion: the word “evident” should be replaced by “obvious”;
  6. pg. 4: the word “Conversely” can be replaced by “consequently”;
  7. All the tables should be inserted into the main text close to their first citation;
  8. pg. 5: table S1 present the Particles size of PtNix/C not the cavity size. Please clarify this aspect;
  9. pg. 6: the sentence “As the Ni to Pt ratio increases, the diffraction peaks shift to higher angles, which is due to the small size of Ni compared to Pt and the resultant lattice contraction of Pt.” should be reformulated. (the shift of the diffraction peaks are due to the variation of the lattice parameters, which varies function of the Ni content of the sample and consequently function of the Ni atomic size);
  10. pg. 8: please increase the resolution of the Figure legend;
  11. 8: please replace “CV patterns” by “CV curves”;
  12. the References should be writhen according to the journal requirements;
  13. pg. 15, Table 2, sample PtNi3/C-AT has an average size of 9 nm and a shell thickness of 5.3 nm. Please clarify this data, since 2Xshell thickness is 10.6 nm;
  14. tables S1, S2, S3, S4 and S5 are supplementary data?
  15. please prepare the Tables according to the journal requirements.

Round 2

Reviewer 1 Report

The authors have corrected the comments and the article can be published.

Reviewer 2 Report

The manuscript has has been revised in accordance with the reviewer comments.

I recommend to accept the paper in present form.